# Sources of Care Stress of Nursing Staff for Patients with Infectious Diseases during the Prevalence of COVID-19: A Case Study of Some Regional Teaching Hospitals in Southern Taiwan

**DOI:** 10.3390/healthcare9040462

**Published:** 2021-04-14

**Authors:** Yichao Huang, Lichen Yu

**Affiliations:** Department of Industrial Management, National Pingtung University of Science and Technology, Pingtung 91201, Taiwan; lijean1225@yahoo.com.tw

**Keywords:** COVID-19, negative pressure isolation ward, care stress, Gaussian curve

## Abstract

(1) Background: The COVID-19 epidemic had caused more than 100 million confirmed cases worldwide by the end of January 2021. The focus of this study was to explore which stress was felt the most by nursing staff in isolation wards in the face of dangerous infectious diseases. (2) Methods: Nursing staff in negative pressure isolation wards were taken as the research objects. The sources of stress were divided into 14 items in three categories, namely, patient care, infection protection, and support system, and the questionnaire results were ranked by a Gaussian curve. (3) Results: Even during the COVID-19 epidemic, nurses in isolation wards still consider that the clinical symptoms of patients in isolation wards cannot be closely tracked as the primary consideration. (4) Conclusions: During the epidemic period, the ability and confidence of nursing staff were strengthened through education and training, and their chances of infection were reduced through comprehensive vaccination and the improvement of protective equipment. In the face of the unstable mood of patients and their families due to isolation, more protective measures should be prepared for nursing staff. In order to relieve the stress, supervisors can adjust the nursing manpower timely according to the difficulty and risk of patient care to reduce the care stress.

## 1. Introduction

Nursing is considered to be a job with high risk, high stress and long working hours [1,2]. In the medical care system, nursing staff play multiple roles and shoulder heavy responsibilities. They are not only the direct caregivers of patients and the coordinators of medical care but also defenders of patients’ life, health and safety. At the same time, they have to assist in performing administrative and quality control work, such as quality control audits, competency advancements, and various evaluations. As nursing is a unique profession, medical teams and the public have high expectations for nursing staff and requirements on the quality of care, and they have to face stress and challenges from different levels, such as the coordination of relationships with physicians and patients’ family members, the management of emergencies, and the risk and stress of infection. Improper handling of incidents can affect patient safety and the quality of medical care. However, nursing staff’s feeling of the stress of nursing work varies with the different working experience and working attributes of different departments. In other words, the more complex a patient’s condition is, the higher the nursing staff’s work stress will be. When nursing staff are faced with failure to provide complete quality of care, they will have care stress [3,4].

In addition to taking care of patients, nursing staff also face the risk of caring for patients with various diseases and of being infected by infectious diseases. In 2003, the world was plunged into unprecedented chaos and panic because of the severe acute respiratory syndrome (SARS) storm, and Taiwan was not spared. Based on the WHO case definition of August 2003, there were 346 confirmed cases with 73 deaths (37 of which were directly due to SARS, and 36 that were SARS-related) in Taiwan [5]. This epidemic caused great impact and trauma to medical care personnel. Due to convenient transportation and the rapid movement of people, it is difficult to prevent and control infectious diseases. Since SARS, Taiwan has continued to experience highly infectious diseases, such as novel H1N1 and H7N9 influenza, the Ebola virus, and Middle East respiratory syndrome coronavirus (MERS-CoV), etc. The COVID-19 epidemic, which began in late 2019, had caused more than 100 million confirmed cases worldwide and more than two million deaths by the end of January 2021. More than 900 people in Taiwan have also been infected, including some medical care personnel caring for patients. During hospitalization, patients with infectious diseases have unstable moods and even violent tendencies due to their condition changes and isolation. Medical staff face the stress of violence and infection while caring for them. This paper used the Gaussian function, which was highly differential and avoided the extreme phenomenon where one person’s decision exceeds the others’ decisions, to calculate the score of the importance of each item. This study aimed to understand the stress of nursing staff in the face of caring for patients with infectious diseases and to find strategies that could help deal with the stress of nursing staff in practice.

## 2. Materials and Methods

In this study, the research motivation was triggered by the research background. After establishing the research theme and direction, relevant literature was collected and analyzed, and the research framework was established. The research questionnaire was established and designed by means of research tools. Questionnaires for nursing staff who met the research conditions were conducted, collected and analyzed. This study mainly took nursing staff currently working in isolation wards as the research objects. Firstly, we studied the literature on nursing work pressure in isolation wards and the impact of COVID-19 on medical staff. Finally, this paper clarified possible sources of the stress for nursing staff in isolation wards based on a number of literature and nursing supervisor interviews. The sources of stress for nursing staff in isolation wards were divided into three categories: patient care, infection protection and support system, which are described below:(1)Patient Care

It was pointed out in the past that nursing staff have complex working environments and interpersonal relationships, and if they cannot reach a consensus with medical staff in the communication process, they will have work stress [6,7]. In addition, when a patient’s condition is critical or the course of a disease changes, it will be regarded as a part of the stress by the nursing staff [8]. Clinical nursing staff are worried about being infected by bacteria, viruses, parasites and infectious diseases when caring for patients. These factors may cause work stress for nursing staff. Nursing staff, based on Nightingale’s duty, believe that these patients should receive appropriate medical care, but they are worried about their own health and safety, thus causing psychological stress in the face of infectious diseases [9]. When patients with infectious diseases are told of their disease, they often cannot accept their disease, just like patients with cancer [10], and they are prone to denial, anger, anxiety and violence [11]. Nursing staff may experience verbal and physical violence from patients [12], therefore, when patients have negative emotions, it is easy to increase the difficulties and work stress of nursing staff in care.(2)Infection Protection

Taiwan was hit hard by SARS in 2003, and COVID-19, which began to spread in late 2019, has caused some medical staff to be infected by patients; therefore, medical units have paid more attention to protective measures for medical staff in the medical process. Although medical institutions began to pay attention to safety protection measures after SARS, the most important defense line should still be the proper use of personal protective equipment and respiratory protective equipment [13,14]. When protective measures are not rigorous enough, there may be vulnerabilities in infection control and epidemic prevention, resulting in cluster infections between medical care personnel and their families [15]. Because some infectious diseases do not show obvious symptoms at the initial stage of the disease, they are usually found after obvious symptoms appear or are confirmed by inspection reports. Cluster infections of nursing staff due to intensive care are reported from time to time, and nursing staff are worried about potential or unknown workplace hazards while caring for patients. In addition, it has been pointed out that as many as 96.2% of nursing staff are extremely afraid of being infected during the care process [16]. Costa et al. (2011) pointed out that nursing staff are more likely to be infected when taking care of patients with infectious diseases [17]. Therefore, the risk of being exposed to infectious diseases as a result of caring for patients is a concern for clinical nursing staff. The loss of the sense of control over the disease, the health of family members, the change of work duties and the isolation due to infectious diseases in the care process will cause the stress of nursing staff in care [9].(3)Support System

The job of nursing staff is challenging and uncertain, and there is no doubt that the job is stressful in the face of illness, death and the demands of patient families on the quality of care. In the medical system, nursing staff are the main force of hospital front-line work and have high work stress in the care work, because they not only need to take care of patients but must also have close contact and communication with their families and medical team. When a patient’s condition changes, coupled with the lack of medical autonomy and supportive working environment for nursing staff, their stress will arise spontaneously [18,19].

Based on the above references, the study found that when dealing with patients with infectious diseases, nursing staff, based on their duties, believe that patients with infectious diseases should receive appropriate medical care, but on the other hand, they worry that they are vulnerable to being infected in a high-risk infection environment and becoming a vector of disease transmission. They worry they do not know enough about the care of the disease to provide complete care, whether they have sufficient capacity to deal with and support patients in serious and critical conditions, and whether their personal safety can be guaranteed. According to the above three categories, 14 sources of stress were worked out through discussions between the head nurses and subordinate nurses of negative pressure isolation wards, as shown in Table 1. The 14 items were made into a questionnaire to be completed by the nursing staff, and each member of the nursing staff ranked the 14 items according to the importance according to their personal experience.

After the questionnaire was completed, this paper calculated the importance of each item using a Gaussian curve. In the previous literature, the Delphi method was used to determine the importance of items [20]. In the Delphi method, the importance of each item is scored by experts. The higher the average score, the more important the item is. In the case of the same average score, the standard deviation and mode of each item will be compared. There may be the same average score among a few items. However, it is relatively rare for many items to have the same average score, and in this case the importance will be evaluated by the average score only. The other methods are to give different weights according to the different importance of items [21], as shown in scales 1–6 in Table 2. In Table 2, m is the ranking of the importance of each item. For the most important item, m = 1, for the second most important item, m = 2, and the rest are similar. If the weight of the most important item is 1, N_p_ represents the number of items, N_t_ represents the number of respondents, and the weight of each item is the weight (*W*_m_), as shown in Table 2.

Scale 7 is the rank-order centroid (ROC) [22], and the score of the importance of item is calculated by Equation (1): (1)$$V_{m}=\frac{1}{N_{p}}\sum_{k=m}^{N_{p}}(\frac{1}{k})$$
where *m* is the ranking of the importance of an item.

In scales 1–4, the difference between the scores of the most and least important items is not high, so the differentiation degree may be insufficient. In scales 5–7, the difference between the scores of the most and least important items is very high, and the differentiation degree is high, but the disadvantage is that when one person considers a certain item as the most important one but others consider it as the least important item, if the score for the most important item is too high, one person’s decision will be more critical than the common decision of the others. In order to avoid this phenomenon, this paper used the Gaussian function to calculate the score of the importance of each item, taking the number of people into consideration. The score is calculated by Equation (2): (2)$$V_{m}=e^{\left(-\frac{{\left(m-1\right)}^{2}}{C}\right)}$$
where *C* is a constant

The score for the most important item: (3)$$V_{1}=e^{\left(-\frac{{\left(1-1\right)}^{2}}{C}\right)}=1$$

The score for the least important item (item N_p_): (4)$$V_{\mathrm{Np}}=e^{\left(-\frac{{\left(N_{p}-1\right)}^{2}}{C}\right)}$$

In order to avoid the circumstance where one person considers an item to be important but others consider it as the least important item, the score of the most important item is set to be the score of the least important item multiplied by (the total number of people—1): $$1=\left(N_{t}-1\right).e^{-(\frac{{\left(N_{p}-1\right)}^{2}}{C})}$$
where *N_t_* is the number of respondents.

Thereby,
C = −(N_p_ − 1)^2^/ln(1/(*N_t_* −1))(5)

Taking the questionnaire with 14 items and 32 respondents as an example, the weights of each scale from Table 2 were as shown in Figure 1. The weight ratio of questions 1 and 14 is listed in Table 3. In scales 1–4, V_1_/V_Np_ was between 3 and 14, and there was no clear distinction. Scales 5–7 showed a clear distinction, but the difference in importance was too great, which may be due to the fact that one person’s decision could negate the others’ decisions. The multiple of Gaussian sequence V_1_/V_Np_ was 31, which was highly differential and avoided the extreme phenomenon where one person’s decision exceeds the others’ decisions. This paper described the importance of a problem using a Gaussian curve, which could consider both the distinction in importance and the number of questionnaire respondents. Therefore, there were no problems, as listed in scales 1–7.

Here, the questionnaire with five items and six respondents was taken as an example. Table 4 shows the importance of each item attached by the six people, wherein 1 represents the most important. According to Equation (5), the following equation can be obtained:C = −(5 − 1) ^2^/ln (1/(6 − 1)) = 9.94

By substituting C = 9.94 into Equation (2), V1 = 1, V2 = 0.904, V3 = 0.669, V4 = 0.404, and V5 = 0.2 could be obtained, respectively. By putting these five values into Table 4, the weight table could be obtained, as shown in Table 5, and the average weight of each item could be obtained to compare their importance. It could be seen from Table 5 that item 3 was the most important item, with the highest average weight of 0.814, while item 5, with a weight of 0.458, was the least important among all items.

## 3. Results

In this study, nursing staff in isolation wards in a number of regional teaching hospitals in Taiwan were selected as the research objects. The COVID-19 epidemic situations are different in varied areas, so nursing staff shoulders diverse responsibilities and faces various risk levels. Therefore, the subject of this study is the nursing staff in isolated wards of hospitals located in varied areas. In the end of 2019, COVID-19 has widely and rapidly spread, resulting in more than 100 million confirmed cases within a year, and the number of subject, which is the nursing staff in the isolation wards of regional hospitals, is in a minority. Consequently, the staff in the negative pressure isolation wards has been coping with such a huge workload in a short time and endured heavy working pressure. By taking the timeliness of the peak period of epidemic prevention into consideration and without increasing any extra burden on nursing staff, this study is based on the theory of central limit theorem, which describes if the number of samples is greater than 30, the sampling distribution will be almost the same as normal distribution. Thus, 32 nursing staff members are randomly chosen as the subjects of this study.

The weight of each item in the questionnaire filled by each respondent was calculated using Equation (2), and the averages are shown in Table 6.

Table 6 shows the weight ranking of the sources of stress for nursing staff in caring for patients with infectious diseases. The attributes of the items with top eight weights were as follows: (1) clinical symptoms of patients in isolation wards cannot be closely tracked; (2) the duration of treatment is extended and the workload becomes heavier; (3) after entering the isolation ward, the nursing staff cannot handle any other patient who is ringing the call bell in time; (4) worrying about being infected while caring for patients with infectious diseases; (5) worrying about whether the current protective equipment can protect you; (6) not being familiar with the samples of infectious diseases to be tested; (7) support from family/colleagues; and (8) care stress resulting from unfamiliarity with the infectious diseases that have not been cared for. In the following sections is a discussion on the first eight sources of stress, so as to understand the causes of these stresses of nursing staff in the process of caring in isolation wards. Supervisors can take the ranking results of the sources as the basis for ward management to reduce the amount of mental disturbance in the caring environment and improve the nurses’ work efficiency and quality of care.

## 4. Discussion

### 4.1. Clinical Symptoms of Patients in Isolation Wards Cannot Be Closely Tracked

The results of this study showed that this attribute ranked first among all attributes. Even at the outbreak of an epidemic, the greatest source of stress for nursing staff is still that the clinical symptoms of patients in isolation wards cannot be closely tracked, which shows that self-work requirements are still the highest professional purpose of nursing staff during their education, which is an admirable spirit. Nursing staff provide 24 h care to patients and are the most direct medical care providers among all medical team members. Nursing staff are exposed to high stress in a patient-centered and service-oriented medical environment. Patients and their families have high expectations and requirements for the quality and results of medical care, but there may be a gap between patients’ expectations and medical results, therefore the relationship between medical care staff and patients can become increasingly tense, and medical disputes can also increase. If the patient’s condition is unstable, with a sharp change in the course of disease progress, or when caring for dying patients, nursing staff may have increased care stress in the face of unfamiliar diseases or a lack of relevant disease care experience. In the event of a sudden outbreak of infectious diseases, nursing staff remain in an isolated environment that cannot be mastered at any time. In particular, nursing staff often have to face emergencies and deal with problems alone due to inadequate support systems and manpower, as well as the fact that physicians cannot deal with patients’ problems immediately. In order to solve the medical care problems of patients, nursing staff have to constantly communicate with the medical team, and the relationship between them can become increasingly tense. When the patients’ problems cannot be dealt with, the psychological stress of nursing staff will arise spontaneously. Therefore, the most important care stress for nursing staff is that the clinical symptoms of patients in isolation wards cannot be closely tracked, resulting in the delayed treatment of patients.

### 4.2. The Duration of Treatment Is Extended and the Workload Becomes Heavier

During the study, the nursing staff said that each patient needs different nursing care, and different types of isolation equipment are required for caring for patients with infectious diseases in different types of isolation, therefore they have to implement correct protective measures after exposing themselves to each patient, such as washing hands and removing isolation clothes. In order to protect themselves from infectious diseases and avoid nosocomial infection, nursing staff must also strictly follow the procedures of putting on and taking off isolation equipment before caring for the next patient. The process of frequently implementing protective measures takes more time than caring for general acute patients, which increases the working time and extends the time of caring for other patients. It is necessary to wear isolation equipment for a long time and to put on and take off protective equipment frequently during work, which increases the workload and finally leads to the stress of nursing staff [23]. The increase in caring time prolongs the working time of nursing staff and increases their workload, which affects the progress of nursing work and the quality of care.

Excessive workloads may cause staff to feel inadequate about their professional knowledge and ability and reduce their sense of competence [24]. Nursing hours are virtually increased by implementing isolation measures and putting on and taking off protective equipment while caring for patients with infectious diseases, causing nursing staff to not have enough time to interact with patients. Due to the implementation of more nursing assessments and non-nursing work, partial nursing measures cannot be fully implemented, which not only affects the holistic and continuous care quality of patients [25] but also increases the caring time and makes nursing staff leave work later, resulting in their physical and mental fatigue. In the long run, it may cause a decline in job satisfaction and eventual resignations. The increase in turnover rate will directly affect the stability of the unit.

### 4.3. After Entering the Isolation Ward, Nursing Staff Cannot Handle Any Other Patient Who Is Ringing the Call Bell in Time

In order to avoid the spread of infectious diseases in the hospital, the hardware of the isolation ward is set to a closed environment. Automatic doors are provided in each area and ward as controls for access to the isolation ward. Different isolation equipment should be selected depending on the type of isolation for patients with infectious diseases due to different patient attributes, and it takes more time to put on and take off the isolation equipment frequently than to care for general acute patients. Such care stress caused by constantly wearing isolation equipment in and out of isolation wards at work is similar to that found in a previous Taiwanese study [24]. Isolation measures must be taken when caring for patients with infectious diseases, and the increased workload makes nursing staff feel overwhelmed, even if they want to provide holistic care and deal with patients’ problems immediately. In addition, the insufficient support system for nursing staff may delay the treatment of patients’ problems, thus affecting the quality of care and reducing the satisfaction of patients’ family members with medical services, which makes nursing staff more worried about the occurrence of medical disputes [3,4]. This study found that when nursing staff enter isolation wards to perform medical care, they cannot keep their mind on the care work because of their concern about the condition of other patients. Nursing is highly professional work. Nursing staff have high self-requirements for the quality of patient care, and each patient has their own care needs. Delays in dealing with patients’ problems will cause work stress, and at the same time, it will also contradict and conflict with the spirit of patient-centered medical care and the emphasis on cross-team medical cooperation.

### 4.4. Worrying about Being Infected while Caring for Patients with Infectious Diseases

In a medical environment, nursing staff are medical workers who directly face and care for patients, and they are more likely to be infected while caring for patients with infectious diseases than other staff members. COVID-19, SARS, tuberculosis, measles, influenza, scabies and other diseases have all caused nosocomial infections in medical institutions. Therefore, comprehensive vaccination has been established as the first line of defense. However, in this study, it was found that if the symptoms were not obvious or the information provided by the patients was insufficient, it was easy to ignore the infectivity of the patients’ diseases. Usually, patients will only be diagnosed when they have symptoms after a period of hospitalization or have been confirmed by a test report, leaving nursing staff as a susceptible group. Nursing staff worry about being infected unexpectedly. Once this happens, medical institutions can pay a great price. In addition to manpower shortages, they may even need to close wards for large-scale environmental cleaning and disinfection before admitting patients again. The infected staff members may have psychological obstacles and must receive long-term health tracking and psychological counseling, so as to help them out of the haze of infection. It can be seen that nursing staff are also concerned about the risk of infection while caring for patients with infectious diseases.

### 4.5. Worrying about whether the Current Protective Equipment Can Provide Protection

Isolation equipment must be selected at different levels depending on the infectivity and route of transmission of the patients’ infectious diseases. Even two layers of protective equipment are needed at work to prevent highly contagious and lethal diseases. Because the process of putting on and taking off isolation equipment is cumbersome and must be operated carefully, nursing staff worry about exposing themselves to the risk of contamination and infection while putting it on and taking it off. Moreover, wearing heavy isolation clothes and gloves to perform medical care work will not only make nursing staff have body discomfort caused by sweating but also affect the flexibility and accuracy of nursing staff in technical operations, as well as indirectly affect the quality of nursing and the patients’ family members’ trust in the nursing staff. In the course of the study, the nursing staff said that they had different isolation equipment depending on the type of isolation of patients with infectious diseases, but in the face of rare infectious diseases, such as novel influenza, measles and MERS-CoV, they worried about whether their protective equipment was correct and safe. However, due to limited manpower, there were no other staff members to help ensure that the protective equipment and procedures used by the nursing staff were correct when admitting patients. Such uncertainty increased the psychological stress of nursing staff and the fear of infection. In addition, in this study, the nursing staff expressed their concern about whether the protective equipment had sufficient effectiveness to protect them. Nursing staff provide direct care for patients and are the majority users of protective equipment and health care materials. As some hospitals are limited in purchasing medical and health care materials, including relevant protective equipment such as N95 masks, protective clothing and masks, due to the factors of the operation system, nursing staff have doubts about the protection of isolation equipment and lack of security in use.

### 4.6. Not Being Familiar with the Samples of Infectious Diseases to Be Tested

A patient who is clinically suspected or confirmed to be infected with a notifiable infectious disease must be notified by the medical institution within the time prescribed by regulations related to notifiable infectious diseases. However, the quality of collecting and sending the samples of patients with infectious diseases is related to the correctness of the test results, which will directly affect the diagnosis of patients with infectious diseases and the subsequent medical care and treatment of infectious diseases. In the course of this study, the nursing staff indicated that different protective measures must be taken according to different routes of infection in clinical care for patients with different infectious diseases, and different types of samples for different infectious diseases should be collected and sent for testing. In some cases, changes to the standard procedures led to the stress of nursing staff over fear of making errors in the operation steps during implementation, and they were confused about the collection of samples to be tested for notifiable infectious diseases. The nursing staff said that some patients were sampled for a second time because the nursing staff had prepared the wrong sampling tube, resulting in the distrust of patients and affecting the timeliness of medical care and patient safety. Such professional disharmony causes the stress of nursing staff [24,26], and they will worry about making mistakes again when encountering similar situations in the future.

### 4.7. Support from Family/Colleagues

Nursing staff are always under high stress in the working environment. An adequate support system will help them to adjust the work stress. McCann (1997) mentioned the influence of family and colleague support in the working environment [27]. The majority of nursing staff are female and they play multiple roles, causing their work stress [6] (McGrath et al., 2003). An adequate support system can improve the performance of nursing staff. In particular, there is a higher risk of infection in the care of patients with infectious diseases than that of ordinary patients, and nursing staff are prone to having negative emotions during the implementation of care, therefore the support from family members is relatively important [28]. Whether a nursing staff member continues to care for patients with infectious diseases or chooses to transfer to other units depends on their families’ recognition of their work. In particular, after marriage, they have to take into account family affairs in addition to work. In case of conflict and unbalance between work and family, the support system will affect the choice of nursing staff to stay in the original unit, transfer to another unit, or leave the workplace. The long-term high-stress environment will increase the turnover of nursing staff [8,23].

For nursing staff, besides family members, their peers also belong to an important support system at work. Especially when they are faced with work stress and difficulties, they are most eager to get support and assistance from their peers and supervisors. Similar research results have also been found abroad [28,29,30]. If nursing staff can get support and care from their supervisors and colleagues at work, share care experiences with each other to face work difficulties and stress, and help each other adapt to and solve problems encountered at work, it will help nursing staff to face work stress with a positive attitude, promote them to develop their expertise in the professional field, and further improve the quality of medical care.

### 4.8. Care Stress Resulting from Unfamiliarity with Infectious Diseases That Have Not Been Cared for Previously

Nursing staff are responsible for caring for patients while facing a variety of different diseases. When faced with diseases and examinations that have not been cared for previously, or when they lack patient care skills, they will have work stress [31]. Due to the convenience of transportation, diseases are no longer limited to certain regions. For example, measles, which was a rare disease in Taiwan in the past 10 years, has been imported from abroad since April 2019 [32]. Medical institutions are all concerned about measles because of its rapid spread. During the process of this study, the nursing staff of the study unit were caring for patients with measles for the first time in their career; they had never been exposed to patients with measles, and they lacked practical experience in caring for the disease. They were not sure whether their antibodies against measles could produce enough protection. In addition, some nursing staff members were infected by patients in the process of care at that time. The nursing staff were worried about being infected or even spreading the disease to their families in the process of care, which caused great stress on the nursing staff in the process of care. Since the outbreak of COVID-19 epidemic in Taiwan in January 2020, there have been continuous cases imported from abroad and cluster infections at home. The nursing staff working in infectious disease words in Taiwan have begun to face more infectious diseases. The rapid change of infectious diseases and changes in clinical conditions have increased the fear of diseases and the stress from the risk of infection for nursing staff.

## 5. Conclusions and Suggestions

Nursing staff have to face the risk and stress of caring for patients with various diseases and of being infected by infectious diseases when caring for patients. This study used a Gaussian curve to calculate the weights of questionnaire items, and took 32 members of nursing staff currently caring for patients in isolation wards as the research objects. A total of 14 stress assessment attributes in caring for patients with infectious diseases were sorted out and summarized according to the literature and interviews. The attributes were ranked by nursing staff according to their importance. The relative weights of the attributes were calculated by the formula according to their importance. During the outbreak of COVID-19, some medical care personnel caring for patients in Taiwan have also been infected. Medical care personnel should be the first to be vaccinated so as to establish the first line of defense against infection. Medical staff’s ability to respond to infectious diseases should be strengthened through on-the-job education and regular epidemic prevention training, so that they will not be caught unprepared in the event of an outbreak, as well as to enhance staff’s confidence and care skills in caring for infectious patients. It is suggested to make posters for the process of sampling and the process of putting on and taking off protective equipment and post them at the operation site, so that staff can easily check and ensure their safety in the process of implementation. In addition, devices that utilize the Internet of Things can be used to overcome the problem of monitoring patients’ conditions. In order to ensure the personal safety of nursing staff, emergency remote controls that can be carried by nursing staff can be used to immediately notify security for help, so as to improve the inadequate support system. In order to improve the quality of care, it is suggested to adjust the allocation of nursing manpower according to the difficulty and risk of patient care so as to reduce the stress of nursing staff in caring for patients with infectious diseases and increase their willingness to care. Only by retaining nursing talents can the quality of care be continuously improved.

## Figures and Tables

**Figure 1 healthcare-09-00462-f001:**
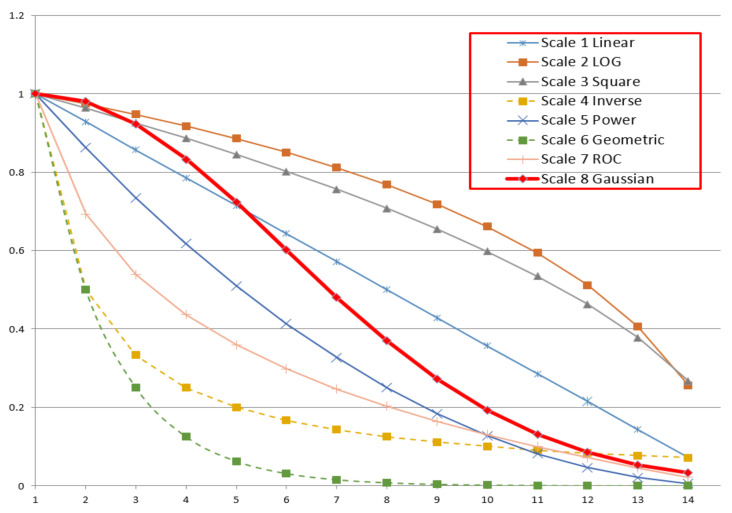
Curves of scales.

**Table 1 healthcare-09-00462-t001:** Evaluation of the care stress attributes of infectious patients.

Category	Item
Patient care	1. Clinical symptoms of patients in isolation wards cannot be closely tracked.
	2. After entering the isolation ward, the nursing staff cannot handle any other patient who is ringing the call bell in time.
	3. The duration of treatment is extended and the workload becomes heavier.
	4. Care stress resulting from unfamiliarity with infectious diseases that have not been cared for previously.
	5. Inconsistency in operating standards causes problems.
	6. Not familiar with the samples of infectious diseases to be tested.
Infection protection	7. Worrying about whether the current protective equipment can protect you.
	8. Worrying about being infected while caring for patients with infectious diseases.
	9. Fear of being contaminated while wearing or taking off protective equipment.
	10. It is inconvenient and inflexible to work in isolation clothes and gloves.
	11. When you are alone with a patient in an isolation ward, you will worry about your own safety.
	12. Fear of spreading an infection to your family.
Support system	13. Support from family/colleagues.
	14. Support from patients’ families and assistance from the medical team.

**Table 2 healthcare-09-00462-t002:** Weight of different scales.

No	Scale	Value	Weight (*W*_m_)
1	Linear (Saaty, 1977)	V_m_ = (N_p_ + 1 − m)	*W*_m_ = V_m_/V_1_
2	Logarithmic	V_m_ = $log_{2}^{\left(N_{p}+1-m+1\right)}$	*W*_m_ = V_m_/V_1_
3	Root Square (Harker and Vargas, 1987)	V_m_ = $\sqrt[{2}]{N_{p}+1-m}$	*W*_m_ = V_m_/V_1_
4	Inverse Linear (Ma and Zheng, 1991)	N_p_/m	*W*_m_ = V_m_/V_1_
5	Power (Harker and Vargas, 1987)	V_m_ = (N_p_ + 1 − m)^2^	*W*_m_ = V_m_/V_1_
6	Geometric (Lootsma, 1989)	V_m_ = $2^{\left(N_{p}-m\right)}$	*W*_m_ = V_m_/V_1_
7	Rank-Order Centroid (ROC)	V_m_ = $\frac{1}{N_{p}}\sum_{k=m}^{N_{p}}\frac{1}{k}$	*W*_m_ = V_m_/V_1_
8	Gaussian Function	V_m_ = $e^{\left(-\frac{{\left(m-1\right)}^{2}}{c}\right)}$ c = −(N_p_ − 1)^2^/ln(1/(N_t_ − 1))	*W*_m_ = V_m_

**Table 3 healthcare-09-00462-t003:** V_1_/V_Np_ of scales.

No	Scale	V_1_/V_Np_
1	Linear (Saaty, 1977)	14
2	Logarithmic	3.91
3	Root Square (Harker and Vargas,1987)	3.74
4	Inverse Linear (Ma and Zheng, 1991)	14
5	Power (Harker and Vargas,1987)	196
6	Geometric (Lootsma, 1989)	8192
7	ROC	45.52
8	Gaussian function	31

**Table 4 healthcare-09-00462-t004:** Response of each item.

Item	Respondent					
	A	B	C	D	E	F
1	5	5	2	2	1	1
2	1	1	5	3	2	3
3	2	3	1	1	4	2
4	4	2	3	5	5	4
5	3	4	4	4	3	5

**Table 5 healthcare-09-00462-t005:** Average weight of each item.

	Respondent						Average	Index
	A	B	C	D	E	F		
1	0.2	0.2	0.904	0.904	1	1	0.713	3
2	1	1	0.2	0.669	0.904	0.669	0.740	2
3	0.904	0.669	1	1	0.404	0.904	0.814	1
4	0.404	0.904	0.669	0.2	0.2	0.404	0.464	4
5	0.669	0.404	0.404	0.404	0.669	0.2	0.458	5

**Table 6 healthcare-09-00462-t006:** Ranking of the weights of stress attributes.

Attribute of Item	Average Weight	Ranking
1. Clinical symptoms of patients in isolation wards cannot be closely tracked.	0.934	1
3. The duration of treatment is extended and the workload becomes heavier.	0.780	2
2. After entering the isolation ward, the nursing staff cannot handle any other patient who is ringing the call bell in time.	0.774	3
8. Worrying about being infected while caring for patients with infectious diseases.	0.716	4
7. Worrying about whether the current protective equipment can protect you.	0.587	5
6. Not familiar with the samples of infectious diseases to be tested.	0.577	6
13. Support from family/colleagues.	0.555	7
4. Care stress resulting from unfamiliarity with infectious diseases that have not been cared for previously.	0.466	8
9. Fear of being contaminated while wearing or taking off protective equipment.	0.448	9
11. When you are alone with a patient in an isolation ward, you will worry about your own safety.	0.433	10
5. Inconsistency in operating standards causes problems.	0.420	11
9. Fear of spreading an infection to your family.	0.244	12
14. Support from patients’ families and assistance from the medical team.	0.200	13
10. It is inconvenient and inflexible to work in isolation clothes and gloves.	0.091	14

## Data Availability

Our data were made available with the submission.

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
