# Peer review of "Sources of Care Stress of Nursing Staff for Patients with Infectious Diseases during the Prevalence of COVID-19: A Case Study of Some Regional Teaching Hospitals in Southern Taiwan"

_healthcare, 2021, doi:10.3390/healthcare9040462_

Round 1
Reviewer 1 Report
Dear authors,
The purpose of this manuscript is to uncover the soucres of care stress of nursing staffs during the prevalence of COVID-19. The research question is interesting and the methodology is well-designed.
As a reviewer, I suggest some of the following comments.
First,I suggest reinforcing the research background for the readers to catch why the methodology is taken in this paper.
Second, I suggest rewriting the Discussion section since it is too heavy and unclear to deliever the insights from the findings.
Author Response
Point 1 :
First, I suggest reinforcing the research background for the readers to catch why the methodology is taken in this paper.
Response 1:
Thank you very much for your kind advice.
The background has been added to Introduction as follow: (line 61-63)
This paper used the Gaussian function, which was highly differential and avoided the extreme phenomenon where one person's decision exceeds the others' decisions, to calculate the score of the importance of each item.
Point 2:
Second, I suggest rewriting the Discussion section since it is too heavy and unclear to deliver the insights from the findings.
Response 2:
Deleted five items of latter section of Discussion. A concise statement was rewritten in conclusion section as follow: (line 441-450)
During the outbreak of COVID-19, some medical care personnel caring for patients have also been infected. Medical care personnel should be the first to be vaccinated so as to establish the first line of defense against infection. Medical staff's ability to respond to infectious diseases should be strengthened through on-the-job education and regular epidemic prevention training, so that they will not be caught unprepared in the event of an outbreak, as well as to enhance staff's confidence and care skills in caring for infectious patients. It is suggested to make posters for the process of sampling and the process of putting on and taking off protective equipment and post them at the operation site, so that staff can easily check and ensure their safety in the process of implementation.
Reviewer 2 Report
The study aimed to understand the stress of nursing staff in the face of caring for patients with infectious diseases and to find strategies that could help deal with the stress of nursing staff in practice. There are a number of questions that arise in the reading of this paper that are reported below.
Abstract:
1/The aim of the study must be justified
Introduction
2/ Lines 60 to 64 should be moved in the Method section.
3/Lines 69 to 71 should be moved in the Introduction section just before the aim of the study.
Method
4/Line 73. The literature search strategy should be more explicit.
5/Line 129: How these 14 items were made. Only about a literature review?
6/ Explain sample size you need to explore an validated your questionnaire.
Author Response
The study aimed to understand the stress of nursing staff in the face of caring for patients with infectious diseases and to find strategies that could help deal with the stress of nursing staff in practice. There are a number of questions that arise in the reading of this paper that are reported below.
Point 1:
Abstract:
- The aim of the study must be justified
Response 1:
Background of abstract is revised as follow:(line 10-13)
The COVID-19 epidemic had caused more than 100 million confirmed cases worldwide by the end of January 2021. The focus of this study was to explore which stress was felt the most by nursing staff in isolation wards in the face of dangerous infectious diseases.
Point 2 :
Introduction
2/ Lines 60 to 64 should be moved in the Method section.
Response 2:
The sentences between line 60 to 64 have been moved to the Method section. (line 67-71)
Point 3 :
3/Lines 69 to 71 should be moved in the Introduction section just before the aim of the study.
Response 3:
The sentences between line 69 to 71 have been moved to the abstract (background). (line 10-13)
Point 4:
Method
4/Line 73. The literature search strategy should be more explicit.
Response 4:
Firstly, we studied the literature on nursing work pressure in isolation wards and the impact of COVID-19 on medical staff. This paper clarified possible sources of the stress for nursing staff in isolation wards based on past literature and nursing supervisor interviews. Finally, the sources of stress for nursing staff in isolation wards were divided into three categories: patient care, infection protection and support system, which are described below. (line 72-78)
Point 5 :
5/Line 129: How these 14 items were made. Only about a literature review?
Response 5:
Firstly, we studied the literature on nursing work pressure in isolation wards and the impact of COVID-19 on medical staff. Finally, this paper clarified possible sources of the stress for nursing staff in isolation wards based on a number of literature and nursing supervisor interviews. (line 72-76)
Point 6:
6/ Explain sample size you need to explore an validated your questionnaire.
Response 6:
Explain sample size in "Result" section:
The COVID-19 epidemic situations are different in varied areas, so nursing staff shoulders diverse responsibilities and faces various risk levels. Therefore, the subject of this study is the nursing staff in isolated wards of hospitals located in varied areas. In the end of 2019, COVID-19 has widely and rapidly spread, resulting in more than 100 million confirmed cases within a year, and the number of subject, which is the nursing staff in the isolation wards of regional hospitals, is in a minority. Consequently, the staff in the negative isolation wards has been coping with such a huge workload in a short time and endured heavy working pressure. By taking the timeliness of the peak period of epidemic prevention into consideration and without increasing any extra burden on nursing staff, this study is based on the theory of central limit theorem, which describes if the number of samples is greater than 30, the sampling distribution will be almost the same as normal distribution. Thus, 32 nursing staff members are randomly chosen as the subjects of this study. (line 218-229)
Reviewer 3 Report
This study is very interesting, namely to analyze the feelings of nurses caused by this new pandemic and understand how much stress they are subjected to and how, possibly, to be able to cope with the problems.
I think the importance of this study lies precisely in going to look for and analyse the stress of healthcare staff, who are too often abandoned and not properly taken into account.
The introductory part of the study, the explanation of the objectives of the study itself and the methods are written very well, precise and punctual.
In my opinion, the part where it is explained how researchers use Gaussian tocalculate the weights of questionnaire items, it isn't very well explained and this makes it a bit difficult to be read.
I would suggest maybe rephrasing some part.
Conclusions and discussion are well written, well explained and in my opinion researchers did a very good source search work.
My only suggestion is to rewrite some the methods part and maybe explain better why they are using that method (gaussian...).
Author Response
Point 1:
This study is very interesting, namely to analyze the feelings of nurses caused by this new pandemic and understand how much stress they are subjected to and how, possibly, to be able to cope with the problems.
I think the importance of this study lies precisely in going to look for and analyse the stress of healthcare staff, who are too often abandoned and not properly taken into account.
The introductory part of the study, the explanation of the objectives of the study itself and the methods are written very well, precise and punctual.
In my opinion, the part where it is explained how researchers use Gaussian to calculate the weights of questionnaire items, it isn't very well explained and this makes it a bit difficult to be read.
I would suggest maybe rephrasing some part.
Conclusions and discussion are well written, well explained and in my opinion researchers did a very good source search work.
Response 1:
Thank you for your kind support.
Point 2:
My only suggestion is to rewrite some the methods part and maybe explain better why they are using that method (gaussian...).
Response 2:
This paper used the Gaussian function, which was highly differential and avoided the extreme phenomenon where one person's decision exceeds the others' decisions, to calculate the score of the importance of each item. This study utilizes Gaussian function as the judgment of weight. The words and formulas between equation (2) and equation (5) in this study explain Gaussian function creates conspicuous weight ratios and the factor of taking the number of subjects into consideration. (line 167-181)
Reviewer 4 Report
This is a very interesting work in the field of nursing and linked to the current pandemic.
I congratulate the authors for the statistical approach. However, I have a suggestion that would add value to the work. Although the variables used are of interest, the temporalisation of events has not been emphasised. I consider it very important to qualify the timing and related measures of restriction if possible in the included studies, as they are variables to be considered that could influence. I recommend this work as a possible model:
Murphy, M., & Moret-Tatay, C. (2021). Personality and Attitudes Confronting Death Awareness During the COVID-19 Outbreak in Italy and Spain. Frontiers in Psychiatry, 12.
Author Response
Point 1:
This is a very interesting work in the field of nursing and linked to the current pandemic.
I congratulate the authors for the statistical approach. However, I have a suggestion that would add value to the work. Although the variables used are of interest, the temporalisation of events has not been emphasised. I consider it very important to qualify the timing and related measures of restriction if possible in the included studies, as they are variables to be considered that could influence. I recommend this work as a possible model:
Murphy, M., & Moret-Tatay, C. (2021). Personality and Attitudes Confronting Death Awareness During the COVID-19 Outbreak in Italy and Spain. Frontiers in Psychiatry, 12.
Response 1:
We added explanation in "Result" section as follow:
The COVID-19 epidemic situations are different in varied areas, so nursing staff shoulders diverse responsibilities and faces various risk levels. Therefore, the subject of this study is the nursing staff in isolated wards of hospitals located in varied areas. In the end of 2019, COVID-19 has widely and rapidly spread, resulting in more than 100 million confirmed cases within a year, and the number of subject, which is the nursing staff in the isolation wards of regional hospitals, is in a minority. Consequently, the staff in the negative pressure isolation wards has been coping with such a huge workload in a short time and endured heavy working pressure. By taking the timeliness of the peak period of epidemic prevention into consideration and without increasing any extra burden on nursing staff, this study is based on the theory of central limit theorem, which describes if the number of samples is greater than 30, the sampling distribution will be almost the same as normal distribution. Thus, 32 nursing staff members are randomly chosen as the subjects of this study. (line 218-229)
Round 2
Reviewer 2 Report
The authors have taken all my comments into account.